rsob.royalsocietypublishing.org

Subject Area:
neuroscience

Keywords:
Huntington's disease, GABAergic signalling, GABA_A receptors, cation–chloride cotransporter

Author for correspondence:
Yijuang Chern
e-mail: bmychern@ibms.sinica.edu.tw

A contribution to the special collection commemorating the 90th anniversary of Academia Sinica.

# Insights into GABA_Aergic system alteration in Huntington's disease

Yi-Ting Hsu[1,2], Ya-Gin Chang[3,4] and Yijuang Chern[1,5]

[1]PhD Program for Translational Medicine, China Medical University and Academia Sinica, Taiwan, Republic of China
[2]Department of Neurology, China Medical University Hospital, Taichung, Taiwan, Republic of China
[3]Institute of Neuroscience, National Yang-Ming University, Taipei, Taiwan, Republic of China
[4]Taiwan International Graduate Program in Interdisciplinary Neuroscience, National Yang-Ming University and Academia Sinica, Taipei, Taiwan, Republic of China
[5]Institute of Biomedical Sciences, Academia Sinica, Taipei, Taiwan, Republic of China

Y-TH, 0000-0002-8552-131X

Huntington's disease (HD) is an autosomal dominant progressive neurodegenerative disease that is characterized by a triad of motor, psychiatric and cognitive impairments. There is still no effective therapy to delay or halt the disease progress. The striatum and cortex are two particularly affected brain regions that exhibit dense reciprocal excitatory glutamate and inhibitory gamma-amino butyric acid (GABA) connections. Imbalance between excitatory and inhibitory signalling is known to greatly affect motor and cognitive processes. Emerging evidence supports the hypothesis that disrupted GABAergic circuits underlie HD pathogenesis. In the present review, we focused on the multiple defects recently found in the GABAergic inhibitory system, including altered GABA level and synthesis, abnormal subunit composition and distribution of GABA_A receptors and aberrant GABA_A receptor-mediated signalling. In particular, the important role of cation–chloride cotransporters (i.e. NKCC1 and KCC2) is discussed. Recent studies also suggest that neuroinflammation contributes significantly to the abnormal GABAergic inhibition in HD. Thus, GABA_A receptors and cation–chloride cotransporters are potential therapeutic targets for HD. Given the limited availability of therapeutic treatments for HD, a better understanding of GABAergic dysfunction in HD could provide novel therapeutic opportunities.

## 1. Introduction

Huntington's disease (HD) is a progressive and autosomal dominant neurodegenerative disorder that is caused by a CAG repeat expansion in exon 1 of the huntingtin (*htt*) gene on the short arm of chromosome 4. This expansion encodes an expanded polyglutamine stretch in the Huntingtin (HTT) protein [1]. The clinical manifestations of HD include movement dysfunction, cognitive decline and psychiatric disorders, with great variation between patients. It has been well established that the age of disease onset is greatly affected by the lengths of the CAG repeats, genetic modifiers and environmental stresses [2–4]. When the CAG repeat number is 40 or higher, nearly full penetrance is expected. The mean age of onset is 40 years, and the illness duration is 15–20 years from the time of symptomatic onset. Neuronal intranuclear inclusions of mutant HTT protein are the characteristic neuropathology of HD, and these inclusions cause extensive cell loss throughout the brain, especially the striatum and cortex.

Gamma-amino butyric acidergic (GABAergic) projecting neurons of the dorsal striatum and glutamatergic pyramidal neurons of the cortex constitute the corticostriatal pathway [5], and these neurons are the most affected neurons in HD [6]. Conversely, GABAergic interneurons in these brain areas are relatively spared [7]. The balance between the excitatory glutamatergic system and the inhibitory GABAergic system is critical for motor and behaviour control.

Dysfunction of this circuit could lead to the development of HD symptoms [8]. The neurons most vulnerable to the toxicity caused by mutant HTT are the medium spiny neurons (MSNs) that project to the globus pallidus externa as part of the indirect basal ganglia pathway [9,10]. The death of these MSNs leads to a disinhibition of the thalamus and hyperactivation of the direct basal ganglia pathway, which subsequently causes involuntary movements (i.e. chorea). As disease progresses, the involvement of cell loss or dysfunction in other regions of the brain, including the thalamus and cerebral cortex, could cause cognitive deficits and mood disorders [11]. No therapy is currently available to delay the onset or slow disease progression for patients with HD.

Dysfunction of the glutamatergic pathway and the subsequent excitotoxicity were investigated extensively in HD [12–14]. Emerging evidence of GABAergic dysfunction was noted in HD. The present manuscript reviews these recent findings on GABAergic dysfunction in HD and discusses the possibility that GABAergic dysfunction may contribute significantly to HD pathogenesis. Further understanding of the abnormal regulation of the important components involved in GABAergic dysfunction may pave the way for the future development of novel therapeutic treatments for HD.

## 2. Overview of GABAergic signalling

Three aspects of the GABAergic system are discussed in this section: (i) the production, release, reuptake and metabolism of GABA; (ii) GABA receptors; and (iii) the actions of GABA (figure 1).

GABA is the principal inhibitory neurotransmitter in adult brains. Both neurons and astrocytes can synthesize GABA and release it to activate GABA receptors on neighbouring neurons. Neuronal GABA is synthesized by two glutamic acid decarboxylases (GAD65 and GAD67) from L-glutamate. Intracellular GABA is transported into synaptic vesicles via the vesicular GABA transporter (VGAT), and then released into the synapse following membrane depolarization [15]. Astrocytic GABA is synthesized by monoamine oxidase B (MAOB) and released by the bestrophin 1 (Best1) anion channel when there is a strong electrochemical gradient that drives GABA efflux [16,17]. Released GABA regulates neural function via binding to GABA receptors, which are localized either pre- or postsynaptically. GABA signals are terminated by reuptake from the synapse into neuron or astrocyte by several membrane-bound GATs. GAT1 is generally considered the primary presynaptic neuronal GABA transporter [18], while GAT3 is localized exclusively on astrocytic processes in the cerebral cortex [19]. In both neurons and astrocytes, GABA is metabolized by GABA transaminase (GABA-T). The glutamate/GABA–glutamine cycle is an important metabolite shuttle between neurons and astrocytes, and it facilitates neurotransmitter homeostasis at GABAergic synapses [20]. Notably, astrocytes take up GABA and metabolize it into α-ketoglutarate and then glutamate via the tricarboxylic acid (TCA) cycle. The glutamate is further converted to glutamine by glutamine synthetase (GS) and then transported to neurons. In neurons, glutamate is regenerated by phosphate-activated glutaminase (PAG) from glutamine and used by GAD as a substrate to produce GABA.

There are two different types of GABA receptors, ionotropic GABA$_A$ receptors and metabotropic GABA$_B$ receptors, which are responsible for fast and slow inhibition, respectively [15]. GABA$_A$ receptors are ligand-gated ion channels that are primarily permeable for chloride ions. These receptors are heteropentameric complexes assembled from 19 different subunits ($\alpha_{1-6}$, $\beta_{1-3}$, $\gamma_{1-3}$, $\delta$, $\varepsilon$, $\theta$, $\pi$ and $\rho_{1-3}$) [21,22]. Most GABA$_A$ receptors exhibit a $2\alpha : 2\beta : 1\gamma$ composition, which determines the subcellular distribution and functional properties. GABA$_A$ receptors containing $\alpha_{1-3}$, $\beta_{2/3}$ and $\gamma_2$ subunits are mainly synaptic, whereas $\alpha_{4-6}$- and $\delta$-containing receptors are mainly peri- or extrasynaptically distributed [23]. Two types of GABA$_A$ receptor-mediated signalling can be discerned, phasic and tonic inhibition [24]. Phasic inhibition is mediated via activation of synaptic GABA$_A$ receptors following brief exposure to a high concentration of GABA released from presynaptic vesicles. Conversely, tonic inhibition is mediated via activation of extrasynaptic GABA$_A$ receptors by a low concentration of ambient GABA. GABA$_B$ receptors are members of the G-protein-coupled receptor family. GABA$_B$ receptors are dimers composed of one R1 subunit (R1a or R1b isoforms) and the R2 subunit, which are located pre- and postsynaptically. GABA$_B$ receptors can transduce signals via intracellular second messengers to modulate the functions of channels and receptors [25,26]. To date, there has been no evidence to suggest that GABAergic signals mediated by GABA$_B$ receptors are altered during HD progression. Therefore, no discussion on GABA$_B$ receptors is included in the present review.

The strength and polarity of GABA$_A$ receptor-mediated neuronal inhibition is determined by intracellular chloride concentrations, which are mainly controlled by two cation–chloride cotransporters: K–Cl cotransporter (KCC2) and Na–K–2Cl cotransporter (NKCC1) [27–29]. During late embryonic and early postnatal stages, GABA induces depolarizing responses that are important for neurite outgrowth, synaptogenesis and neural plasticity. This response occurs because NKCC1 is highly expressed in immature neurons and enables significant chloride influx and a high concentration of intracellular chloride. Activation of GABA$_A$ receptors thus causes the efflux of chloride in immature neurons and evokes a depolarizing signal. As neurons mature, NKCC1 is downregulated, and KCC2 is upregulated, and both channels contribute to intracellular chloride concentration regulation and the switching of GABA signals from depolarizing to hyperpolarizing.

## 3. Role of GABA$_A$ergic signalling in mice and humans with HD

Accumulating evidence demonstrates that the GABAergic system, especially GABA$_A$ receptor-mediated signalling (table 1), is altered in the brains of several mouse models of HD, such as R6/2 [41] and R6/1 mice [42], N171-82Q [43], YAC 128 [44] and BACHD transgenic mice [45], Hdh knock-in mice [46], conditional HD mice [35], and HD patients. Note that a more focused review about HD-related changes in the expression of GABA$_A$ receptors (GABA$_A$Rs) has been published recently [47]. Therefore, a dysregulated GABAergic system is an authentic symptom of HD, and it may contribute significantly to HD pathogenesis. Because the striatum and cortex are the two most affected brain areas in HD, alteration of the GABA$_A$ergic system was actively investigated in these two brain areas.

rsob.royalsocietypublishing.org Open Biol. 8: 180165

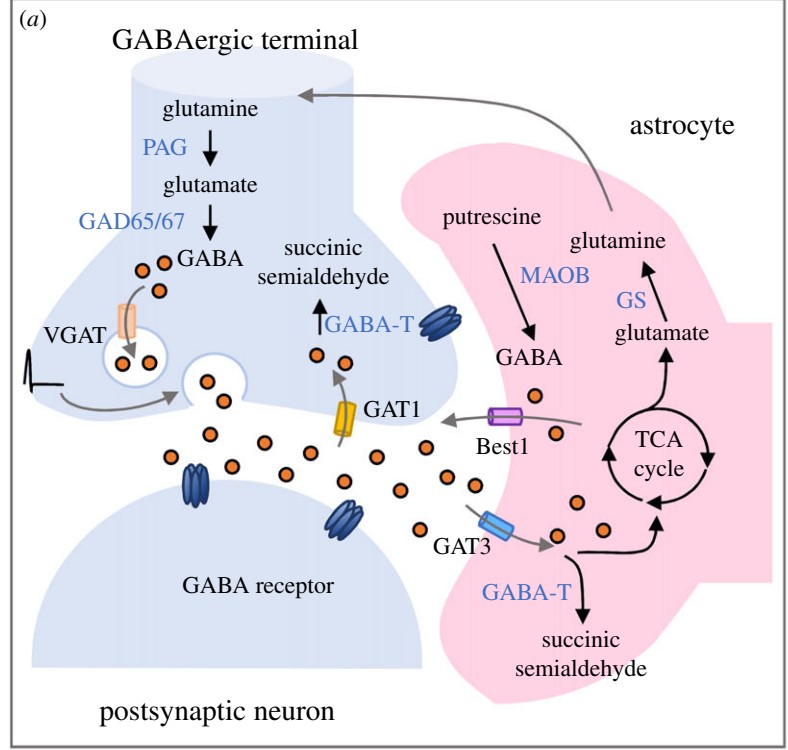

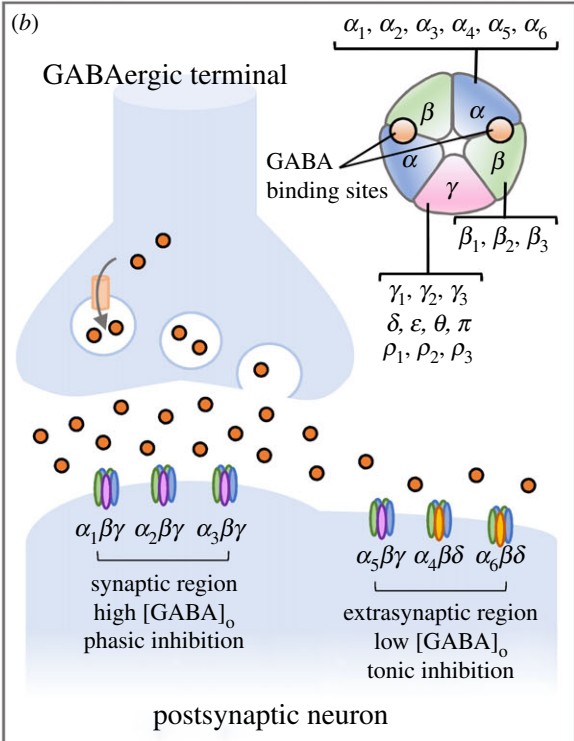

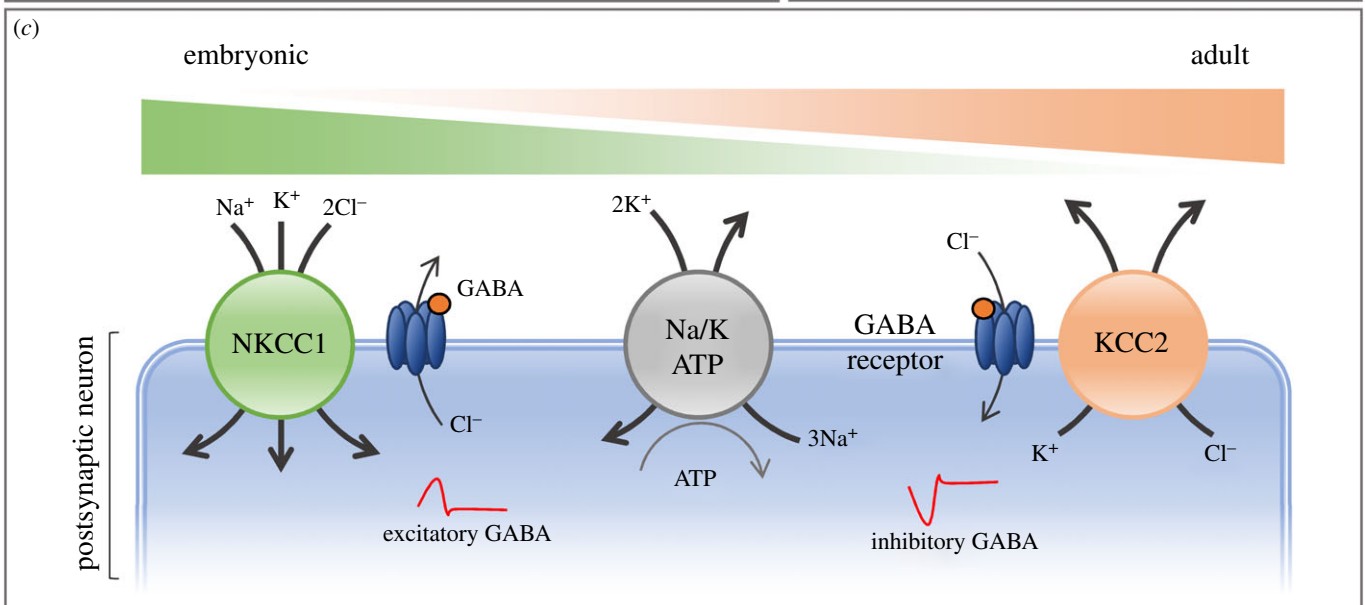

**Figure 1.** An overview of the γ-aminobutyric acid (GABA) signalling system. (a) GABA homeostasis is regulated by neurons and astrocytes. GABA is synthesized by GAD65/67 from glutamate in neurons, while astrocytic GABA is synthesized through MAOB. The release of GABA is mediated by membrane depolarization in neurons and Best1 in astrocytes. The reuptake of GABA is mediated through GAT1 in neurons and GAT3 in astrocytes. The metabolism of GABA is mediated by GABA-T in neurons and astrocytes. The reuptake of GABA in astrocytes is further transformed into glutamine via the TCA cycle and glutamine synthetase (GS). The glutamine is then transported to neurons and converted to glutamate for regeneration of GABA. (b) GABA$_A$ receptors are heteropentameric complexes assembled from 19 different subunits. The compositions of different subunits determines the subcellular distributions and functional properties of the receptors. Phasic inhibition is mediated via the activation of synaptic GABA$_A$ receptors following brief exposure to a high concentration of extracellular GABA. Tonic inhibition is mediated via the activation of extrasynaptic GABA$_A$ receptors by a low concentration of ambient GABA. (c) The excitatory or inhibitory response of GABA is driven by the chloride gradient across cell membranes, which can be determined via two cation–chloride cotransporters (NKCC1 and KCC2). The high expression of NKCC1 during the developmental stage maintains higher intracellular [Cl$^-$] via chloride influx to the cell. The activation of GABA$_A$ receptors at an early developmental stage results in an outward flow of chloride and an excitatory GABAergic response. As neurons mature, the high expression of KCC2 maintains lower intracellular [Cl$^-$] via chloride efflux out of the cell. The activation of GABA$_A$ receptors on mature neurons results in the inward flow of chloride and an inhibitory GABAergic response.

## 3.1. Alterations in GABA levels and metabolism

Earlier studies using postmortem brain tissues from HD patients revealed that HD patients have a lower GABA content in the caudate/putamen and cortex than non-HD subjects [48–51]. It remains elusive whether HD patients have lower activity of GAD, an enzyme responsible for the production of GABA [52,53]. R6/2 and R6/1 mice have lower levels of GAD67 mRNA in the cortex and striatum, respectively, than their littermate controls [30,54]. Neurons differentiated from HD induced pluripotent stem cells (iPSCs) also contain fewer GAD67 transcripts than control

**Table 1.** Changes in GABA$_A$ receptors and related molecules in Huntington's disease. ChAT-IN, choline acetyltransferase positive-interneuron; GAD, glutamic acid decarboxylase; IPSC, inhibitory postsynaptic currents; KCC2, K−Cl cotransporter; mo, months; MSN, medium-sized spiny projection neuron; NKCC1, Na−K−2Cl cotransporter; p't, patient; PPV-IN, parvalbumin-interneuron; VGAT, vesicular GABA transporter; wks, weeks.

| | disease stage | mRNA expression[a] | protein expression[a] | functional[b] |
|---|---|---|---|---|
| cortex | late stage | ↓α$_1$ (R6/2 mice, 12 wks; N171-82Q mice, 16 wks; prefrontal cortex of HD p't)<br>↔α$_2$, α$_3$ (R6/2 mice, 12 wks)<br>↓α$_2$ (*Hdh*$^{150Q}$ mice, 15 mo; prefrontal cortex of HD p't)<br>↓α$_4$ (R6/2 mice, 12 wks)<br>↓δ (R6/2 mice, 12 wks; N171-82Q mice, 16 wks; *Hdh*$^{150Q}$ mice, 15 mo; prefrontal cortex of HD p't)<br>↓GAD67 (R6/2 mice, 12wks)<br>↓KCC2 (*Hdh*$^{150Q}$ mice, 15 mo) | ↓δ (R6/2 mice, 12 wks)<br>↓KCC2 (R6/2 mice, 12 wks; N171-82Q mice, 16wks)<br>↑NKCC1 (R6/2 mice, 12 wks) | ↓IPSC (pyramidal neurons, R6/2 mice, 80 days; N171-82Q mice, 4 mo; BACHD mice, 6 mo; conditional HD mice, 6 mo)<br>↑IPSC (pyramidal neurons, YAC128 mice, 12 mo; Hdh (CAG)150 mice, 12 mo) |
| | early stage | ↔α$_{1−3}$ (R6/2 mice, 7 wks)<br>↓α$_4$ (R6/2 mice, 7 wks)<br>↓δ (R6/2 mice, 7 wks) | | ↑IPSC (pyramidal neurons, R6/2 mice, 40 days; YAC128 mice, 6 mo)<br>↔IPSC (pyramidal neurons, N171-82Q mice, 1−2 mo; BACHD mice, 3 mo) |
| striatum | late stage | ↓α$_2$ (R6/2 mice, 12 wks; *Hdh*$^{150Q}$ mice,15 mo; R6/1 mice, 6 mo)<br>↑α$_3$ (R6/2 mice, 12 wks; R6/1 mice, 6 mo)<br>↓α$_4$ (R6/2 mice, 12 wks; R6/1 mice, 6 mo)<br>↑α$_5$ (YAC128 mice, 12 mo; R6/1 mice, 6 mo)<br>↔β$_{1−3}$ (R6/1 mice, 6 mo)<br>↑β$_3$ (YAC128 mice, 12 mo)<br>↑γ$_1$ (R6/1 mice, 6 mo)<br>↔γ$_2$ (R6/1 mice, 6 mo)<br>↓δ (R6/2 mice, 12 wks; N171-82Q, 16wks; *Hdh*$^{150Q}$ mice, 15 mo; R6/1 mice, 6 mo; caudate of HD p't)<br>↑δ (YAC128 mice, 12 mo)<br>↓GAD67 (R6/1 mice, 6 mo) | ↑α$_1$ (MSN, R6/1 mice, 6 mo; *Hdh*$^{111Q}$, 8 mo)<br>↓α$_1$ (PV-IN, R6/1 mice, 6 mo; putamen of HD p't)<br>↓α$_2$ (MSN, PV-IN, R6/1 mice, 6 mo)<br>↑α$_3$ (MSN, ChAT-IN, R6/1 mice, 6 mo)<br>↑α$_5$ (R6/1 mice, 6 mo; *Hdh*$^{111Q}$, 8 mo)<br>↔β$_{2−3}$ (R6/1 mice, 6mo)<br>↓β$_{2/3}$ (N171-82Q, 4 mo; putamen of HD p't)<br>↔γ$_2$ (R6/1 mice, 6 mo)<br>↓γ$_2$ (putamen of HD p't)<br>↓δ (R6/2 mice, 12 wks; R6/1 mice, 6 mo)<br>↔GAD65/67 (R6/2 mice, 12 wks, R6/1 mice, 6 mo)<br>↑GAD65 (*Hdh*$^{111Q}$, 8 mo)<br>↔VGAT (R6/1 mice, 6 mo)<br>↔NL2, or gephyrin (R6/1 mice, 6 mo)<br>↓KCC2 (R6/2 mice, 12 wks; N171-82Q mice, 16 wks)<br>↑NKCC1 (R6/2 mice, 12 wks; *Hdh*$^{150Q}$ mice, 15 mo; GFAP-HD mice, 18−24 mo) | ↑IPSC (D2-MSN, R6/2 mice, greater than 60 days; MSN, Hdh(CAG)150 mice, 12 mo; YAC128 mice, 12 mo)<br>↓IPSC (MSN, N171-82Q mice, 4mo)<br>↓tonic current (D2-MSN, R6/2 mice, greater than 60 days; MSN, Q175 mice, greater than 1 yr) |

(*Continued.*)

| disease stage | mRNA expression[a] | protein expression[a] | functional[b] |
|---|---|---|---|
| early stage | $\leftrightarrow\alpha_{1-3}$ (R6/2 mice, 7 wks) | $\leftrightarrow\alpha_{1,3,5}$ (R6/1 mice, 2 mo) | $\leftrightarrow$IPSC (MSN, N171-82Q mice, 1–2 mo; BACHD, 3 mo) |
| | $\downarrow\alpha_4$ (R6/2 mice, 7 wks) | $\uparrow\alpha_2$ (MSN, PV-IN, R6/1 mice, 2 mo) | $\uparrow$IPSC (MSN, YAC128 mice, 1 mo) |
| | $\downarrow\alpha_5$ (YAC128 mice, 6 mo) | $\uparrow\beta_2$ (R6/1 mice, 2 mo) | |
| | $\downarrow\beta_3$ (YAC128 mice, 6 mo) | $\leftrightarrow\beta_3$ (R6/1 mice, 2 mo) | |
| | $\downarrow\delta$ (R6/2 mice, 7 wks) | $\leftrightarrow$GAD65/67, VGAT (R6/1 mice, 2 mo) | |
| | $\uparrow\delta$ (YAC128 mice, 6 mo) | $\leftrightarrow$NL2, or gephyrin (R6/1 mice, 2mo) | |
| | | $\leftrightarrow$NKCC1 (R6/2 mice, 7 wks) | |

[a]mRNA and protein expressions of the indicated molecules are summarized from the following references [30–34].
[b]Electrophysiological studies in the striatal and cortical neurons are summarized from the following references [33,35–40].

iPSCs [55]. In addition, a recent study of metabolic profiling in R6/2 mice at a symptomatic stage revealed significant decreases in GABA synthesis in the cortex and striatum using $^{13}$C labelling and mass spectrometry [56]. Overall, these studies suggest a trend of reduced GABA in HD. Whether other components involved in GABA metabolism, such as GABA transporters (GAT1, GAT3), enzymes for GABA degradation (e.g. GABA-T) or channels for glia GABA release (Best1), are affected during HD progression is not known.

## 3.2. Alterations in the subunit composition and distribution of GABA_A receptors

When compared with that of non-HD subjects, a significant loss of GABA_AR binding sites in the caudate of HD patients has been reported [31,57–59]. In particular, the levels of $\alpha$ and/or $\gamma$ subunits that constitute the benzodiazepine-binding sites of GABA_AR are reduced. Changes in the protein and mRNA levels of GABA_A receptor subunits were also observed in the brains of mice and humans with HD [32–34,54,60,61]. Immunohistochemistry staining revealed that the putamen of HD patients contained lower levels of $\alpha_1$, $\beta_{2,3}$ and $\gamma_2$, but not $\alpha_3$ subunits than non-HD subjects [60]. Consistently, we had reported a significant decrease in the transcripts of GABA_AR $\alpha_1$, $\alpha_2$ and $\delta$, but not $\alpha_3$ mRNA in the postmortem prefrontal cortex of HD patients. A trend towards lower levels of $\alpha_2$ and $\delta$ subunit transcripts in the postmortem caudate of HD patients was also found. Different HD mouse models (R6/2, N171-82Q and Hdh CAG knock-in mice) recapitulated these findings at the manifest stage [32]. Reduction of $\alpha_4$ and $\delta$ subunits, which are located mainly perisynaptically and extrasynaptically, can be observed as early as the premanifest stage in the cortex and striatum of HD mice. The $\alpha_4\beta\delta$ subunit-containing GABA_ARs mainly mediate tonic conductance in adult cortical pyramidal cells and striatal medium spiny neurons [23,36]. The reduced cortical and striatal $\alpha_4$ and $\delta$ subunits may result in poor GABAergic tonic inhibition in the early stages of HD, and this likely precedes the appearance of overt symptoms. Interestingly, another group reported that the levels of $\alpha_1$, $\alpha_3$ and $\alpha_5$ subunit mRNAs and proteins are increased in the striatum at the manifest stage in R6/1 mice, and the levels of the $\delta$ subunit are decreased. No change in the levels of these subunits was observed before the onset of motor symptoms in R6/1

mice [54]. These authors further demonstrated changes in GABA_AR subunits in different subtypes of striatal neurons using immunohistochemical staining. They found that the expression level of the $\alpha_1$ subunit increased in MSNs and decreased in parvalbumin (PV)-expressing interneurons. Moreover, the $\alpha_2$ subunit in PVs and MSNs was decreased, while the $\alpha_3$ subunit was increased in the choline acetyltransferase (ChAT)-expressing interneurons at the manifest stage in R6/1 mice. In an HD mouse model (N171-82Q) where mHTT is expressed mainly in neurons, a significant decrease in the surface expression of GABA_AR $\beta_{2/3}$ subunits at the manifest stage was reported. This defect occurs because of the disrupted GABA_AR trafficking but there is no change in total amount [34]. No change in the mRNA or protein expression level of total $\beta_2$, $\beta_3$ or $\gamma_2$ subunits was found in R6/1 mice [54]. No major changes were found in the expression levels of neuroligin or gephyrin proteins, which are involved in the targeting and clustering of GABA_A receptors at the postsynaptic membrane in R6/1 mice [54]. Rosas-Arellano *et al.* [61] recently reported an altered expression of peri- or extrasynaptically distributed GABA_A tonic subunits ($\alpha_5$, $\beta_3$, $\delta$ and $\rho_2$) in the D2 dopamine receptor (D2R)-enriched neostriatal pathway, and their relocalization into the synaptic cleft during the early stage in YAC128 mice. These authors suggested that the intrasynaptic localization of GABA_A tonic subunits was a cellular strategy to compensate for the inferior GABA environment in the HD neostriatum and facilitate the GABA-mediated inhibition. The aforementioned changes in subunit composition and distribution of GABA_A receptors suggest that the GABA-mediated phasic or tonic inhibition may be altered.

GABAergic transmission in cortical and striatal neurons during HD progression has been investigated using electrophysiological approaches in several HD mouse models (R6/2, N171-82Q, YAC128, BACHD, Hdh CAG knock-in mice and conditional HD mice) [34,35,37–40,62]. Both increases and decreases in GABA_A receptor-mediated synaptic currents in cortical pyramidal neurons and striatal neurons have been reported in different HD mouse models. Importantly, the GABA_A receptor-mediated tonic chloride current in striatal neurons is consistently reduced in two HD mouse models (R6/2 and Q175) [39,40]. Reduced tonic GABA current is observed in D2R-expressing MSNs in R6/2 mice. Because the striatal tonic GABA_A currents have been proposed to have a neuroprotective effect against excitotoxicity [63], the

reduced tonic GABA$_A$ currents may contribute to the elevated susceptibility of the D2R-expressing MSNs in HD. Moreover, deficits in GABA-mediated cortical inhibition have been reported in premanifest and early HD patients measured by a transcranial magnetic stimulation approach. The impaired cortical inhibition in HD patients is likely to associate with disease severity, psychiatric performance and neurocognitive disturbances [64]. Taken together, these studies provide strong evidence to support that alterations in GABA$_A$ receptor-mediated tonic and phasic inhibitory transmission occur in HD and may contribute to HD pathogenesis.

### 3.3. Is GABA an inhibitory or excitatory neurotransmitter?

The reversal potential for GABA$_A$-mediated currents is controlled by the two chloride cotransporters, KCC2 and NKCC1. In the cortex and striatum of different HD mouse models, expression of KCC2 and an interacting protein, the brain-type creatine kinase (CKB), is reduced [32]. The interaction between KCC2 and CKB in HD brains is also lower than in their littermate controls. Because the activity of KCC2 is positively controlled by CKB [65,66] and the expression and activity of CKB are reduced in HD [67,68], the function of KCC2 in HD is likely lower, which may subsequently reduce GABA$_A$-mediated inhibitory function. Conversely, NKCC1 expression in the striatum of several HD mouse models (R6/2 mice, N171-82Q and Hdh CAG knock-in mice) is abnormally elevated at the symptomatic stage (Y-T Hsu, Y-G Chang, Y-C Li, K-Y Wang, H-M Chen, D-J Lee, C-H Tsai, C-C Lien, Y Chern 2018, personal communication). Consistently, a depolarized GABAergic response in the striatal neurons of R6/2 mice was observed by a gramicidin-perforated patch-clamp recording, which may reduce the driving force or even reverse the chloride flow following activation of GABA$_A$ receptors. Under this impaired condition, stimulation of GABAergic neurons in the striatum of HD mice might become excitatory.

Moreover, Dargaei et al. [69] have shown that alterations in the cation–chloride cotransporters also occurred in the hippocampus of two HD mouse models (R6/2 and YAC128). These authors reported that reduced KCC2 expression and an increase in NKCC1 expression converted the GABA-mediated stimulation from inhibitory to excitatory in the hippocampus. These studies collectively support the hypothesis that abnormally reduced GABAergic inhibition underlies HD pathogenesis.

Accumulating evidence also suggested that the upregulation of NKCC1 and a downregulation of KCC2 level in epileptogenic regions (such as the hippocampus) increase the susceptibility of developing seizures in rodents and humans [70,71]. In line with this concept, R6/2 mice with anomalous levels of NKCC1 and KCC2 in the hippocampus are more susceptible to the development of seizures [72]. R6/2 mice, a well-established early onset and rapidly progressive transgenic mouse model carrying exon 1 of the HD gene with approximately 130 CAG repeats [41], may capture certain features of juvenile-onset HD patients. Notably, seizures were present in 30–40% of juvenile HD patients and were more likely to occur with earlier disease onset and large CAG expansions (greater than 80 CAG repeats) [73,74]. In contrast to the R6 lines of HD mice, seizures are not present in another early onset HD mouse model, N171-82Q mice, which carry a cDNA fragment encoding an N-terminal fragment of HTT

with 82 glutamines and express mHTT only in neurons [43]. Full length and knock-in HD mouse models, which capture more features of adult-onset HD patients, do not have seizures [75]. Seizure is also infrequent or virtually absent in adult-onset HD patients [74]. These discrepancies regarding seizure occurrence between different mouse models and juvenile- and adult-onset HD are worth further investigation, especially on the involvement of GABAergic dysfunction.

## 4. Molecular mechanism(s) underlying the abnormal GABA$_A$ergic system in HD

In HD, misfolded mutant HTT proteins engage in aberrant interactions with multiple cellular components to cause synaptic dysfunction and the degeneration of specific neurons in the brain [76,77]. The proposed mechanisms of HD pathogenesis include abnormal protein aggregation and clearance, transcriptional deregulation, impaired axonal transport and vesicle trafficking, mitochondrial toxicity, energy imbalance, oxidative stress, glutamate excitotoxicity and neuroinflammation [77–79]. These toxic effects can be mediated through cell-autonomous or cell non-autonomous mechanisms. The potential mechanisms underlying the GABAergic abnormalities are summarized in figure 2.

### 4.1. Molecular mechanisms underlying the mutant HTT-mediated suppression of GABA$_A$ receptors and their downstream signalling molecules

Earlier studies have shown that HTT interacts with many important transcriptional activators and repressors [80,81]. Thus, the loss of normal HTT function due to the poly-Q stretch-mediated changes in protein properties or cellular location of HTT is known to dysregulate the global transcriptional profile in HD brains [80–82]. For example, mutant HTT inhibits the transcription of genes containing the neuron-restrictive silencer factor (NRSF) by failing to interact with repressor element-1 silencing transcription factor (REST) and forming the nuclear corepressor complex at the RE1/NRSE nuclear site [81]. In addition, mutant HTT inhibits the transcriptional activator specificity protein 1 (Sp1) binding to DNA and causes the transcriptional inhibition [80]. Notably, the promoters of GABA$_A$R subunits contain NRSE- and Sp1-binding sites [83]. The KCC2 gene also includes two RE1/NRSE sites that flank its transcription start site [84]. Therefore, mutant HTT might interfere with the transcriptional machinery to reduce the expression levels of KCC2 and GABA$_A$ receptor subunits in HD brains.

HTT also interacts with Huntingtin-associated protein 1 (HAP1), which binds with GABA$_A$R and KIF5 (kinesin motor protein 5) for transport to the synapses [85]. The polyQ-expanded mutant HTT disrupts the HAP1–KIF5 complex, impairs the trafficking of GABA$_A$R to the synapse, and reduces the synaptic inhibition in HD brains [34].

It remains to be determined whether mutant HTT interacts with cation–chloride cotransporters directly or indirectly to affect their function or expression. Two proteomic studies reported a potential association between KCC2 and HTT because the KCC2 protein is highly enriched in the HTT proteome [86,87]. The interaction of HTT and KCC2 was demonstrated in the hippocampus of wild-type and HD mice (R6/2) using coimmunoprecipitation assays. No such interaction between

rsob.royalsocietypublishing.org   Open Biol. 8: 180165

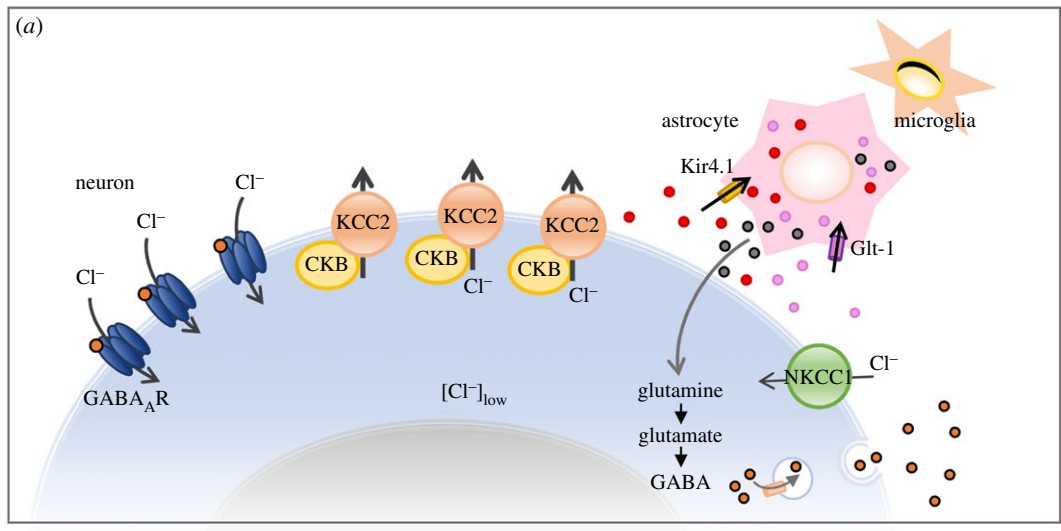

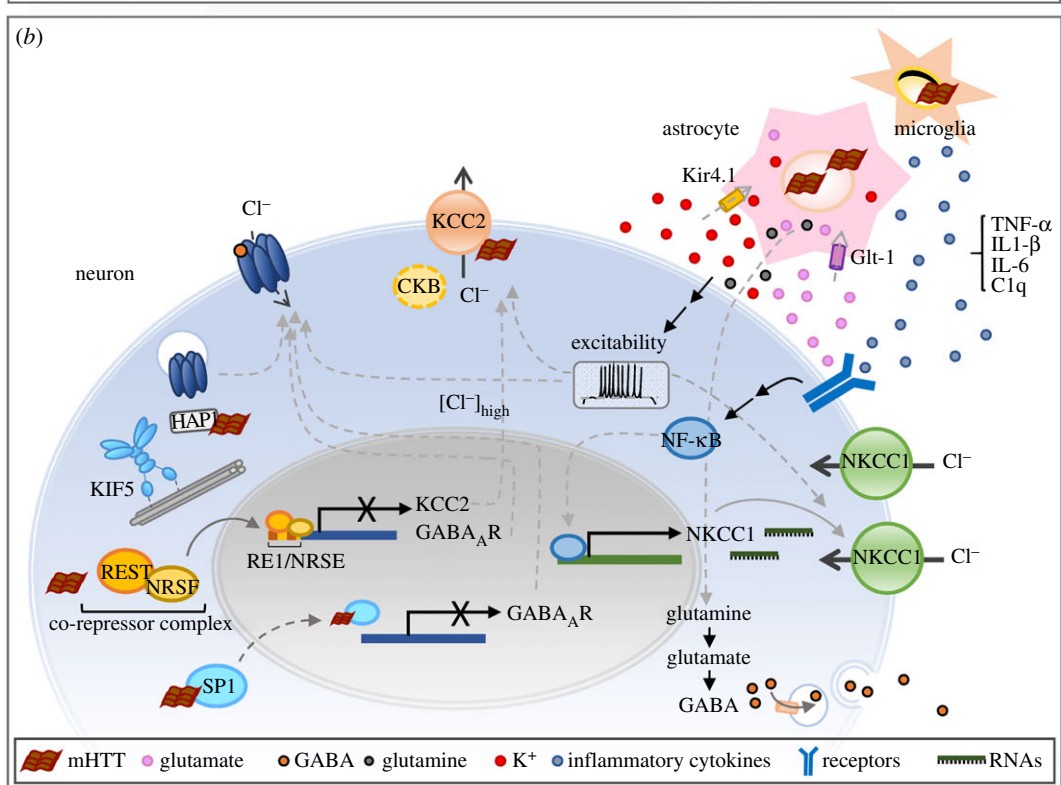

**Figure 2.** Molecular mechanism(s) underlying the abnormal GABA$_A$ergic system in HD. (*a*) In the normal condition, adult neurons express high KCC2 and few NKCC1 to maintain the lower intracellular chloride concentration, which results in an inward flow of chloride when GABA$_A$ receptors are activated. Astrocytes function normally for the homeostasis of glutamate, potassium and glutamate/GABA-glutamine cycle. (*b*) In Huntington's disease, reduced GABA$_A$ receptor-mediated neuronal inhibition is associated with enhanced NKCC1 expression and a decreased expression in KCC2 and membrane localized GABA$_A$ receptors. The dysregulated GABA$_A$ergic system might be caused by mutant HTT, excitotoxicity, neuroinflammation or other factors. Mutant HTT in neurons alters the transcription of genes (*GABA$_A$R* and *KCC2*) through interactions with transcriptional activators (SP1) and repressors (REST/NRSF). Mutant HTT in neurons also disrupts the intracellular trafficking of GABA$_A$Rs to the cellular membrane. HD astrocytes have impaired homeostasis of extracellular potassium/glutamate (due to deficits of astrocytic Kir4.1 channel and glutamate transporters, Glt-1) and cause neuronal excitability, which might be related to the changes of KCC2, NKCC1 and GABA$_A$R. The activity of KCC2 could be affected through its interacting proteins, such as CKB and mHTT. Neuroinflammation, which is evoked by the interaction of HD astrocyte and microglia, enhances NKCC1 expression in neurons at the transcriptional level through an NF-κB-dependent pathway. HD astrocytes also have compromised astrocytic metabolism of glutamate/GABA−glutamine cycle that contributes to lower GABA synthesis.

NKCC1 and HTT was reported [69]. Dargaei *et al.* [69] suggested that KCC2 may be sequestered into mutant HTT inclusions, which greatly interfere with the function and availability of KCC2. Another possibility is that HTT might affect the function of KCC2 by interfering with KCC2-interacting proteins, such as CKB (an enzyme involved in energy homeostasis). Consistent with this hypothesis, reduced CKB expression contributes to neuronal dysfunction in HD pathogenesis [68,88]. This reduction is very important because KCC2 is positively regulated by

CKB via a protein−protein interaction [65,66]. In HD neurons, reduced interaction between KCC2 and CKB was demonstrated by a proximity ligation assay [32].

## 4.2. Enhanced neuroinflammation contributes to the altered GABAergic signalling

Previous studies suggest that inflammatory mediators may affect GABAergic signalling via altering the expression of

rsob.royalsocietypublishing.org Open Biol. 8: 180165

cation–chloride cotransporters, NKCC1 and/or KCC2, to affect chloride homeostasis in models of neuropathic pain [89,90].

Reactive gliosis (i.e. accumulation of activated astrocytes and microglia) and neuroinflammation have been observed in the brains of mice and patients with HD, and are known to contribute to neuronal dysfunction. Several reports have indicated that mutant HTT-expressing astrocytes led to the cell death of neurons [91] and activation of nuclear factor (NF)-κB-mediated inflammatory response [92]. Emerging evidence indicates that microglial activation is also observed in the brains of HD mouse models and patients [93–96]. Abnormally activated microglia and astrocytes result in increased production of inflammatory mediators in HD brains, such as IL1-β, IL-6 and TNF-α [79,92–94,97]. A recent study reported that activated microglia induce neurotoxic reactive astrocytes by secreting IL-1α, TNF and C1q, and reactive astrocytes cause the neuronal dysfunction in neurodegenerative disorders [98]. Expression of mutant HTT in astrocytes led to enhanced NKCC1 expression in neurons at the transcriptional level through an NF-κB-dependent pathway (Y-T Hsu, Y-G Chang, Y-C Li, K-Y Wang, H-M Chen, D-J Lee, C-H Tsai, C-C Lien, Y Chern 2018, personal communication). Consistent with the importance of NF-κB, both the mouse and human *NKCC1* promoters have NF-κB binding sites in their promoter region [99–101]. In addition, two earlier studies showed that the activity and/or protein expression of NKCC1 was positively regulated by NF-κB [102,103]. The aberrant activation of NF-κB-mediated inflammation in mutant HTT-expressing astrocytes likely alters neuronal functions via the cytokine-mediated upregulation of NKCC1 in neurons and changes neuronal responses to GABAergic stimuli.

Notably, neuroinflammation and the GABA neurotransmitter system are reciprocally regulated in the brain (reviewed in [104,105]). Specifically, neuroinflammation induces changes in the GABA neurotransmitter system, such as reduced GABA$_A$ receptor subunit expression, while activation of GABA$_A$ receptors likely antagonizes inflammation. TNF-α, a proinflammatory cytokine, induces a downregulation of the surface expression of GABA$_A$Rs containing $\alpha_1$, $\alpha_2$, $\beta_{2/3}$ and $\gamma_2$ subunits and a decrease in inhibitory synaptic strength in a cellular model of hippocampal neuron culture [106]. The same group further demonstrated that protein phosphatase 1-dependent trafficking of GABA$_A$Rs was involved in the TNF-α evoked downregulation of GABAergic neurotransmission [107]. Upregulation of TNF-α also negatively impacts the expression of GABA$_A$R $\alpha_2$ subunit mRNA and thus decreases the presynaptic inhibition in the dorsal root ganglion in a rat experimental neuropathic pain model [108]. Conversely, blockade of central GABA$_A$Rs in mice by a GABA$_A$R antagonist increased both the basal and restraint stress-induced plasma IL-6 levels [109]. Inhibition of GABA$_A$R activation by picrotoxin increased the nuclear translocation of NF-κB in acute hippocampal slice preparations [110]. Collectively, neuroinflammation weakens the inhibitory synaptic strength in neurons, at least partly, through the reduction of GABA$_A$Rs. The reduced expression and function of GABA$_A$Rs may further increase inflammatory responses. It remains elusive whether the same mechanism occurs in the inflammatory environment in HD brains.

### 4.3. Other potential mechanisms

Astrocytic factor(s) other than neuroinflammation may also contribute to the impaired GABAergic signalling in HD

brains. Activated astrocytes in HD have defects that affect their supportive functions to neurons. For example, HD astrocytes express fewer glutamate transporters than wild-type astrocytes and thus are deficient in astrocytic glutamate homeostasis [56,111] and have abnormal extracellular potassium homeostasis due to astrocytic Kir4.1 channel deficits [112]. The reduced astrocytic glutamate transporters may contribute to glutamate toxicity. Interestingly, excitotoxicity is known to negatively regulate the expression of KCC2 and GABA$_A$ receptor subunits [113,114] and might also affect GABAergic signalling in HD. In addition, Tong *et al.* [112] reported that Kir4.1 deficiency in astrocytes evokes an elevation in striatal extracellular potassium and causes MSN hyperexcitability in two different HD mouse models (R6/2 and Q175). Another study revealed that disturbed astrocyte-mediated potassium buffering caused hyperactivity of neuronal NKCC1 in ammonia-induced neurotoxicity [115]. Thus, hyperexcitability resulting from deficiency of astrocytic Kir4.1 might have also contributed to neuronal NKCC1 upregulation and altered GABAergic signalling in HD brains. Moreover, Skotte *et al.* [56] reported decreased astrocytic glutamine release and compromised astrocytic metabolism of glutamate–GABA–glutamine cycling, which resulted in lower GABA synthesis in the R6/2 mouse model. Thus, astrocytes and their interaction with neurons in HD brains play important roles in GABAergic signalling.

## 5. Potential therapeutic implications in HD

### 5.1. Modulating the GABA$_A$ receptor as a therapeutic target

In view of the presently discovered HD-related deficit in the GABA system, the question arises whether HD patients can benefit from drugs that stimulate the GABA system (figure 3*a*). HD patients suffer from motor abnormalities and non-motor symptoms, including cognitive deficits, psychiatric symptoms, sleep disturbance, irritability, anxiety, depression and an increased incidence of seizures [74,77,116,117]. Seizures are a well-established part of juvenile HD but no more prevalent in adult-onset HD than in the general population [73,74,118]. Several pharmacological compounds can enhance inhibitory GABAergic neurotransmission by targeting GABA$_A$R and thereby producing sedative, anxiolytic, anticonvulsant and muscle-relaxant effects. A recent study demonstrated that zolpidem, a GABA$_A$R modulator that enhances GABA inhibition mainly via the $\alpha_1$-containing GABA$_A$ receptors, corrected sleep disturbance and electroencephalographic abnormalities in symptomatic HD mice (R6/2) [119]. Alprazolam, a benzodiazepine-activating GABA receptor, reversed the dysregulated circadian rhythms and improved cognitive performance of HD mice (R6/2) [120]. In addition, progesterone, a positive modulator of GABA$_A$R, significantly reversed the behavioural impairment in a 3-nitropropionic acid (3-NP)-induced HD rat model [121]. Apart from modulating the activity of the GABAergic system by interfering directly with the receptor, pharmacological agents can also interfere with synaptic GABA concentrations. Tiagabine, a drug that specifically blocks the GABA transporter (GAT1) to increase synaptic GABA level, was found to improve motor performance and extend survival in N171-82Q and R6/2 mice [122]. It is also worth evaluating whether vigabatrin, a

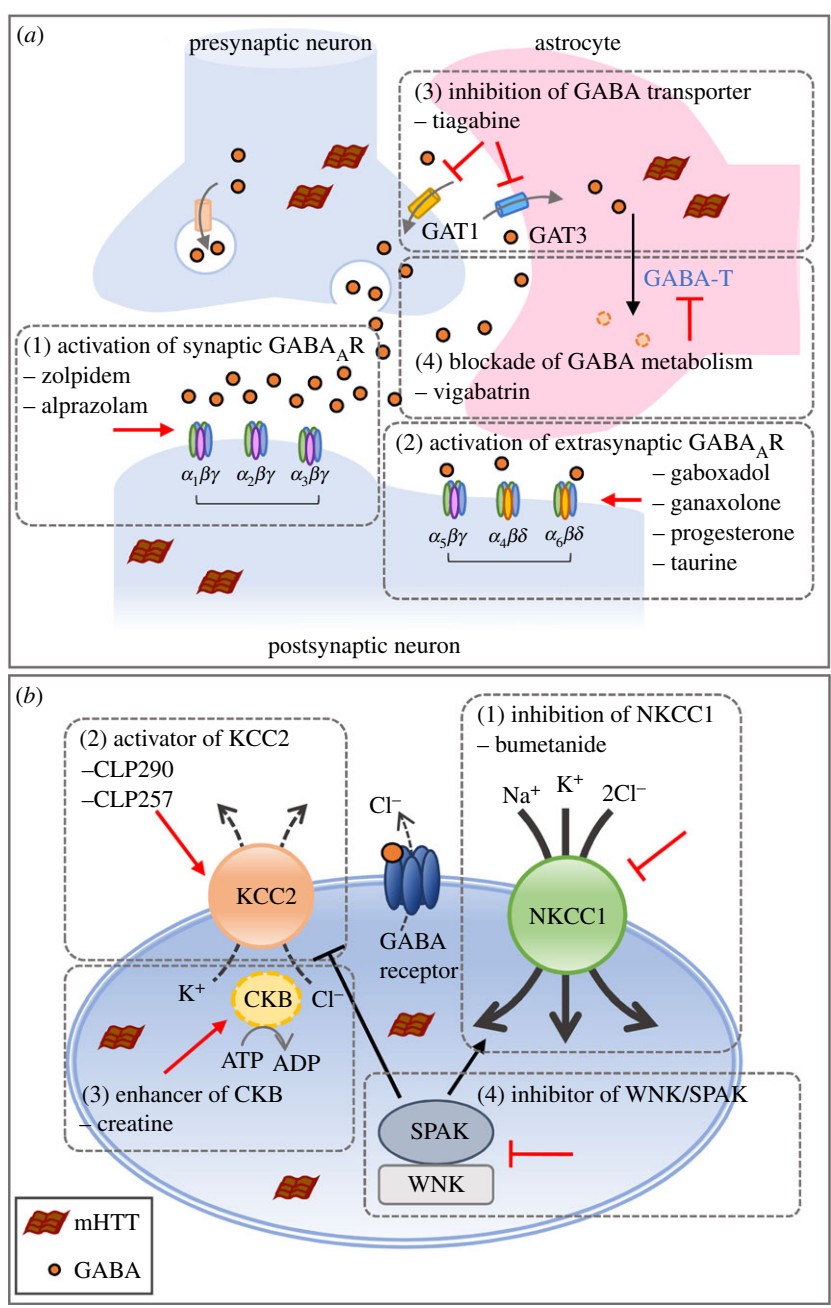

**Figure 3.** Strategy to target (*a*) GABA$_A$R and (*b*) cation–chloride cotransporters as potential therapeutic avenues. (*a*) The GABAergic system is influenced directly by agents that (1) target synaptic GABA$_A$R, (2) increase tonic GABA current or interfere with synaptic GABA concentrations via a reduction of GABA reuptake (3), and (4) block GABA metabolism. (*b*) GABA$_A$R-mediated signalling in HD neurons is depolarizing due to the high intracellular chloride concentration caused by high NKCC1 expression and low KCC2 expression. Rescuing the function of cation–chloride cotransporters can occur via (1) inhibition of NKCC1 activity using bumetanide, (2, 3) increase in KCC2 function using a KCC2 activator or CKB enhancer, and (4) inhibitors of WNK/SPAK kinases.

GABA-T inhibitor that blocks GABA catabolism in neurons and astrocytes [123], plays a role in the compromised astrocytic glutamate–GABA–glutamine cycling [56]. Interestingly, taurine exerted GABA$_A$ agonistic and antioxidant activities in a 3-NP HD model and improved locomotor deficits and increased GABA levels [124].

However, several early studies failed to provide the expected benefits of GABA analogues in slowing disease progression in HD patients [125–127]. For example, gaboxadol, an agonist for the extrasynaptic δ-containing GABA$_A$ receptor, failed to improve the decline in cognitive and motor functions of five HD patients during a short two-week trial, but it caused side effects at the maximal dose [125]. Interestingly, although treatment with muscimol (a potent agonist of GABA receptors)

did not improve motor or cognitive deficits in 10 HD patients, it did ameliorate chorea in the most severely hyperkinetic patient [126]. The therapeutic failure of GABA stimulation in early clinical trials does not argue against the importance of GABA-ergic deficits in HD pathogenesis. The alteration of GABAergic circuits plays a primary role or is a compensatory response to excitotoxicity, and it may contribute to HD by disrupting the balance between the excitation and inhibition systems and the overall functions of neuronal circuits. Because the subunits of the GABA$_A$ receptor are brain region- or neuron subtype-specific, the choice of drugs may have distinct effects on the brain region or neuronal population targeted [128–130]. For example, the expression of GABA$_A$R subunits is differentially altered in MSNs and other striatal interneurons in HD

[54,60]. The early involvement of D2-expressing MSNs can cause chorea [131], while dysfunctional PV-expressing interneurons can cause dystonia in HD patients [132]. Specific alteration in neuronal populations and receptor-subtypes during HD progression needs to be taken into consideration when treating the dysfunction of GABAergic circuitry. Notably, striatal tonic inhibition mediated by the δ-containing GABA_ARs may have neuroprotective effects against excitotoxicity in the adult striatum [63]. Because the reductions in δ-containing GABA_ARs and tonic GABA currents in D2-expressing MSNs have been observed in early HD [32,39,40,54,61], it would be of great interest to evaluate the effects of several available compounds, such as alphaxalone and ganaxolone [133], that target δ-containing GABA_ARs, in animal models of HD.

## 5.2. Modulation of chloride homeostasis via cation–chloride cotransporters

Emerging evidence suggests that chloride homeostasis is a therapeutic target for HD. Pharmacological agents that target cation–chloride cotransporters (i.e. NKCC1 or KCC2) therefore might be used to treat HD (figure 3b). Of note, dysregulation of cation–chloride cotransporters and GABA polarity was associated with several neuropsychiatric disorders [70,134–139] (reviewed in [27,140]). Such abnormal excitatory GABA_A receptor neurotransmission can be rescued by bumetanide, an NKCC1 inhibitor that decreases intracellular chloride concentration. Bumetanide is an FDA-approved diuretic agent that has been used in the clinic. It attenuates many neurological and psychiatric disorders in preclinical studies and some clinical trials for traumatic brain injury, seizure, chronic pain, cerebral infarction, Down syndrome, schizophrenia, fragile X syndrome and autism (reviewed in [141]). Daily intraperitoneal injections of bumetanide also restored the impaired motor function of HD mice (R6/2, Y-T Hsu, Y-G Chang, Y-C Li, K-Y Wang, H-M Chen, D-J Lee, C-H Tsai, C-C Lien, Y Chern 2018, personal communication). The effect of bumetanide is likely to be mediated by NKCC1 because genetic ablation of NKCC1 in the striatum also rescued the motor deficits in R6/2 mice (Y-T Hsu, Y-G Chang, Y-C Li, K-Y Wang, H-M Chen, D-J Lee, C-H Tsai, C-C Lien, Y Chern 2018, personal communication). This study uncovered a previously unrecognized depolarizing or excitatory action of GABA in the aberrant motor control in HD. In addition, chronic treatment with bumetanide also improved the impaired memory in R6/2 mice [69], supporting the importance of NKCC1 in HD pathogenesis. Owing to the poor ability of bumetanide to pass through the blood–brain barrier, further optimization of bumetanide and other NKCC1 inhibitors is warranted [142,143].

Disruption of KCC2 function is detrimental to inhibitory transmission and agents to activate KCC2 function would be beneficial in HD. However, no agonist of KCC2 has been described until very recently [144,145]. A new KCC2 agonist (CLP290) has been shown to facilitate functional recovery after spinal cord injury [145]. It would be of great interest to evaluate the effect of KCC2 agonists on HD progression. Another KCC2 activator, CLP257, was found to increase the cell surface expression of KCC2 in a rat model of neuropathic pain [146]. Post-translational modification of KCC2 by kinases may modulate the function of KCC2. The WNK/SPAK kinase complex, composed of WNK (with no lysine) and SPAK (SPS1-related proline/alanine-rich kinase), is known to phosphorylate and stimulate NKCC1 or inhibit KCC2 [147]. Thus, compounds that inhibit WNK/SPAK kinases will result in KCC2 activation and NKCC1 inhibition. Some compounds have been noted as potential inhibitors of WNK/SPAK kinases and need to be further tested for their effects on cation–chloride cotransporters [148–150]. An alternative mechanism to activate KCC2 is manipulation of its interacting proteins (e.g. CKB [65,66]). Because CKB could activate the function of KCC2 [65,66], CKB enhancers may increase the function of KCC2. In HD, reduced expression and activity of CKB is associated with motor deficits and hearing impairment [68,88]. Enhancing CKB activity by creatine supplements ameliorated the motor deficits and hearing impairment of HD mice. It is worthwhile to further investigate the interaction of KCC2 and CKB in GABAergic neurotransmission and motor deficits in HD.

The depolarizing GABA action with altered expression levels of NKCC1 or KCC2 is associated with neuroinflammation in HD brains [32,69]. Blockade of TNF-α using Xpro1595 (a dominant negative inhibitor of soluble TNF-α) [151] *in vivo* led to significant beneficial effects on disease progression in HD mice [152] and reduced the expression of NKCC1 (Y-T Hsu, Y-G Chang, Y-C Li, K-Y Wang, H-M Chen, D-J Lee, C-H Tsai, C-C Lien, Y Chern 2018, personal communication). It would be of great interest to test the effect of other anti-inflammatory agents [153] on the function and expression of NKCC1 and GABAergic inhibition. Neuroinflammation is implicated in most neurodegenerative diseases, including Alzheimer's disease and Parkinson's disease [154,155], and the interaction of cation–chloride cotransporters and neuroinflammation in GABAergic neurotransmission may also play a critical role in other neurodegenerative diseases.

## 6. Concluding remark

Although tremendous understanding of HD pathogenesis has been achieved in recent years, there is still no effective therapy to prevent or halt disease progression. A global characterization of excitatory and inhibitory neurotransmission in HD brains is critical, and it may provide the molecular insight needed to develop novel therapeutic treatments for HD. In this review, we focused on alterations of GABAergic inhibitory systems, which precede the appearance of overt symptoms in HD. Emerging evidence has revealed alterations in various components of the GABA_A-ergic system, including GABA levels, GAD activity, phasic or tonic GABA currents, the distribution and composition of GABA_AR and cation–chloride cotransporters. Some of these alterations are caused by mutant HTT, neuroinflammation, excitotoxicity and other factors that remain elusive. Altered GABAergic signalling may lead to an imbalance between excitation and inhibition, which underlies the motor and cognitive impairments in HD. In summary, GABA_AR and cation–chloride cotransporters are important therapeutic targets for HD.

Data accessibility. This article has no additional data.
Authors' contributions. Y.T.H., Y.G.C. and Y.C. drafted and revised the manuscript. All authors gave final approval before submission.
Competing interests. We declare we have no competing interests.
Funding. This study was supported by a grant from China Medical University Hospital (CRS-106-060) to Y.T.H.

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
