## [Reviewer comments · Open Biology]

Review History

RSOB-18-0165.R0 (Original submission)

Review form: Reviewer 1

Recommendation

Accept with minor revision (please list in comments)

Are each of the following suitable for general readers?

- a) **Title**
Yes
- b) **Summary**
Yes
- c) **Introduction**
Yes

Is the length of the paper justified?

Yes

Should the paper be seen by a specialist statistical reviewer?

No

Is it clear how to make all supporting data available?

Not Applicable

Is the supplementary material necessary; and if so is it adequate and clear?

Not Applicable

Do you have any ethical concerns with this paper?

No

Comments to the Author

This manuscript is a clear and detailed review describing the current research showing changes in the GABAergic system linked to HD. Compared to the glutamatergic system, the GABAergic system has been for the most part overlooked during the last decades. This is a timely and useful review of our recent knowledge in the field. Note that a more focused review, not cited in this MS, has been published recently (PMID: 29464851).

Minor revisions:

1-Fig 1b displays the heteropentameric complex formed by α , β , γ , ... subunits with a circle illustrating the GABA binding site at the interface between 1 α and 1 β subunit. There are 2 binding sites on such a complex (at the 2 α / β interfaces). Both of them (or none) must be illustrated.

2-Fig. 1b and and Fig. 3a suggest that GABAARs are on spine heads, this is not true. The 2 figures should be changed accordingly.

3-page 8 chapter 3.3: In this chapter it is said that expression of NKCC1 is elevated at the symptomatic stage while the function of KCC2 is lower. This should lead to seizure susceptibility. Epilepsy has been reported in mouse models of HD but is not a hallmark of adult-onset of HD in human (see point 6 below). These discrepancies (changes in chloride homeostasis leading or not to epilepsy, animal models vs human HD....) should be discussed.

4-page 8: "neuroinflammation" ?

5-Page 9 : "...because the KCC2 encoding gene, Slc 12A5, is highly enriched..." should be "...because the KCC2 gene product, Slc 12A5, is highly enriched..." or "...because the KCC2 protein is highly enriched..."

6-Page 11: It is stated that "HD patients suffer from abnormalities.....and an increased incidence of seizures". While epilepsy is common in the juvenile form of HD, it is infrequent or virtually absent in the adult-onset form (e.g. PMID:23124580). Actually, it has been shown that the mutation carrier with an adult onset are less prone to seizure as compared to a non-carrier group. This should be discussed.

7-Page 32, Fig. 2 legend "...through interacting with.." should be "...through interactions with.."

- “disrupts the trafficking of GABAARs to synaptic membrane”. Should be : “disrupts the intracellular trafficking of GABAARs to cellular membrane”.

Review form: Reviewer 2

Recommendation

Accept as is

Are each of the following suitable for general readers?

- a) **Title**
Yes
- b) **Summary**
Yes
- c) **Introduction**
Yes

Is the length of the paper justified?

Yes

Should the paper be seen by a specialist statistical reviewer?

No

Is it clear how to make all supporting data available?

Not Applicable

Is the supplementary material necessary; and if so is it adequate and clear?

Not Applicable

Do you have any ethical concerns with this paper?

No

Comments to the Author

Excellent and clear review. I have no concern regarding this interesting manuscript.

Decision letter (RSOB-18-0165.R0)

17-Oct-2018

Dear Dr Chern

We are pleased to inform you that your manuscript RSOB-18-0165 entitled "Insights into GABAergic system alteration in Huntington's disease" has been accepted by the Editor for publication in Open Biology. The reviewer(s) have recommended publication, but also suggest

some minor revisions to your manuscript. Therefore, we invite you to respond to the reviewer(s)' comments and revise your manuscript.

Please submit the revised version of your manuscript within 14 days. If you do not think you will be able to meet this date please let us know immediately and we can extend this deadline for you.

- 1) A text file of the manuscript (doc, txt, rtf or tex), including the references, tables (including captions) and figure captions. Please remove any tracked changes from the text before submission. PDF files are not an accepted format for the "Main Document".
- 2) A separate electronic file of each figure (tiff, EPS or print-quality PDF preferred). The format should be produced directly from original creation package, or original software format. Please note that PowerPoint files are not accepted.
- 3) Electronic supplementary material: this should be contained in a separate file from the main text and meet our ESM criteria (see <http://royalsocietypublishing.org/instructions-authors#question5>). All supplementary materials accompanying an accepted article will be treated as in their final form. They will be published alongside the paper on the journal website and posted on the online figshare repository. Files on figshare will be made available approximately one week before the accompanying article so that the supplementary material can be attributed a unique DOI.

Online supplementary material will also carry the title and description provided during submission, so please ensure these are accurate and informative. Note that the Royal Society will not edit or typeset supplementary material and it will be hosted as provided. Please ensure that the supplementary material includes the paper details (authors, title, journal name, article DOI). Your article DOI will be 10.1098/rsob.2016[last 4 digits of e.g. 10.1098/rsob.20160049].

- 4) A media summary: a short non-technical summary (up to 100 words) of the key findings/importance of your manuscript. Please try to write in simple English, avoid jargon, explain the importance of the topic, outline the main implications and describe why this topic is newsworthy.

Images

Data-Sharing

It is a condition of publication that data supporting your paper are made available. Data should be made available either in the electronic supplementary material or through an appropriate repository. Details of how to access data should be included in your paper. Please see <http://royalsocietypublishing.org/site/authors/policy.xhtml#question6> for more details.

Data accessibility section

Sincerely,

The Open Biology Team
<mailto:openbiology@royalsociety.org>

ditage Insights by clicking on the following link: <https://www.surveymonkey.com/r/author-perspectives-on-academic-publishing-royal-society>

This should take no more than 15 minutes and you will have the opportunity to enter a prize draw. We hope these results will provide us with valuable insights we can use to improve our service.

Reviewer(s)' Comments to Author:

Referee: 1

Comments to the Author(s)

This manuscript is a clear and detailed review describing the current research showing changes in the GABAergic system linked to HD. Compared to the glutamatergic system, the GABAergic system has been for the most part overlooked during the last decades. This is a timely and useful review of our recent knowledge in the field. Note that a more focused review, not cited in this MS, has been published recently (PMID: 29464851).

minor revisions:

1-Fig 1b displays the heteropentameric complex formed by α , β , γ , δ , ϵ subunits with a circle illustrating the GABA binding site at the interface between 1 α and 1 β subunit. There are 2 binding sites on such a complex (at the 2 α/β interfaces). Both of them (or none) must be illustrated.

2-Fig. 1b and and Fig. 3a suggest that GABAARs are on spine heads, this is not true. The 2 figures should be changed accordingly.

3-page 8 chapter 3.3: In this chapter it is said that expression of NKCC1 is elevated at the symptomatic stage while the function of KCC2 is lower. This should lead to seizure susceptibility. Epilepsy has been reported in mouse models of HD but is not a hallmark of adult-onset of HD in human (see point 6 below). These discrepancies (changes in chloride homeostasis leading or not to epilepsy, animal models vs human HD....) should be discussed.

4-page 8: "neuroinflammation" ?

5-Page 9 : "...because the KCC2 encoding gene, Slc 12A5, is highly enriched..." should be "...because the KCC2 gene product, Slc 12A5, is highly enriched..." or "...because the KCC2 protein is highly enriched..."

6-Page 11: It is stated that "HD patients suffer from abnormalities.....and an increased incidence of seizures". While epilepsy is common in the juvenile form of HD, it is infrequent or virtually absent in the adult-onset form (e.g. PMID:23124580). Actually, it has been shown that the mutation carrier with an adult onset are less prone to seizure as compared to a non-carrier group. This should be discussed.

7-Page 32, Fig. 2 legend "...through interacting with.." should be "...through interactions with.." - "disrupts the trafficking of GABAARs to synaptic membrane". Should be : "disrupts the intracellular trafficking of GABAARs to cellular membrane".

Referee: 2

Comments to the Author(s)

Excellent and clear review. I have no concern regarding this interesting manuscript.

Author's Response to Decision Letter for (RSOB-18-0165.R0)

See Appendix A.

Decision letter (RSOB-18-0165.R1)

30-Oct-2018

Dear Dr Chern

We are pleased to inform you that your manuscript entitled "Insights into GABAergic system alteration in Huntington's disease" has been accepted by the Editor for publication in Open Biology.

Sincerely,

The Open Biology Team
mailto: openbiology@royalsociety.org

Appendix A

中央研究院
生物醫學科學研究所

ACADEMIA SINICA
INSTITUTE OF BIOMEDICAL SCIENCES
Taipei, Taiwan

Yijiang Chern, Ph. D.
Director, Department of International Affairs
Distinguished Research Fellow
Institute of Biomedical Sciences
Academia Sinica
Taipei, 11529, Taiwan
Tel: 886-2-26523913
Fax: 886-2-27829143
bmychern@ibms.sinica.edu.tw

Oct. 24, 2018

Dear Prof. Glover:

Enclosed is a revision of our manuscript titled “Insights into GABAergic system alteration in Huntington’s disease” (manuscript no: RSOB-18-0165). We wish to thank you and the reviewers for your helpful comments. We attempted to address each comment and question either in the body of the revised manuscript or in this letter. Our responses to the specific comments of Reviewers are delineated below.

Referee: 1

This manuscript is a clear and detailed review describing the current research showing changes in the GABAergic system linked to HD. Compared to the glutamatergic system, the GABAergic system has been for the most part overlooked during the last decades. This is a timely and useful review of our recent knowledge in the field. Note that a more focused review, not cited in this MS, has been published recently (PMID: 29464851).

RESPONSE: As the Reviewer pointed out, dysregulation of the GABAergic system is a timely issue. We have cited the indicated review (PMID: 29464851) in the revised manuscript and would like to thank the Reviewer for this suggestion (page 5).

minor revisions:

1. Fig 1b displays the heteropentameric complex formed by α , β , γ , ... subunits with a circle illustrating the GABA binding site at the interface between 1 α and 1 β subunit. There are 2 binding sites on such a complex (at the 2 α/β interfaces). Both of them (or none) must be illustrated.

RESPONSE: We have modified Fig. 1b as recommended in the revised manuscript, and would like to thank the Reviewer for point out this mistake.

2. Fig. 1b and and Fig. 3a suggest that GABAARs are on spine heads, this is not true. The 2 figures should be changed accordingly.

RESPONSE: We have modified Figs. 1b and 3a as recommended in the revised manuscript, and would like to thank the Reviewer for this suggestion.

3. page 8 chapter 3.3: In this chapter it is said that expression of NKCC1 is elevated at the symptomatic stage while the function of KCC2 is lower. This should lead to seizure susceptibility. Epilepsy has been reported in mouse models of HD but is not a hallmark of adult-onset of HD in human (see point 6 below). These discrepancies (changes in chloride homeostasis leading or not to epilepsy, animal models vs human HD....) should be discussed.

RESPONSE: As suggested, we added a new paragraph in the revised manuscript to discussion this issue (page 8, the last paragraph).

4. page 8: “neuroflamation” ?

RESPONSE: We corrected this typo into “neuroinflammation” in the revised manuscript (page 9).

5. Page 9 : “...because the KCC2 encoding gene, Slc 12A5, is highly enriched...” should be “...because the KCC2 gene product, Slc 12A5, is highly enriched...” or “...because the KCC2 protein is highly enriched...”

RESPONSE: We modified the indicated sentence in the revised manuscript (page 10) as suggested.

6. Page 11: It is stated that “HD patients suffer from abnormalities.....and an increased incidence of seizures”. While epilepsy is common in the juvenile form of HD, it is infrequent or virtually absent in the adult-onset form (e.g. PMID:23124580). Actually, it has been shown that the mutation carrier with an adult onset are less prone to seizure as compared to a non-carrier group. This should be discussed.

RESPONSE: We agree with the Reviewer and had included a paragraph to the revised manuscript (page 8, the last paragraph; page 12).

7-Page 32, Fig. 2 legend “....through interacting with..” should be “....through interactions with..”

- “.....disrupts the trafficking of GABAARs to synaptic membrane”. Should be : “.....disrupts the intracellular trafficking of GABAARs to cellular membrane”.

RESPONSE: We corrected the indicated sentences as recommended in the legend of Figure 2.

Referee: 2

Excellent and clear review. I have no concern regarding this interesting manuscript

RESPONSE: We would like to thank the reviewer for this comment.

As we have discussed over emails in April, four review manuscripts (including this one) from the NPAS (http://npas.programs.sinica.edu.tw/index_en.html) at Academia Sinica would be submitted to the *OPEN BIOLOGY* and published together to celebrate the 90th anniversary of Academia Sinica. Dr. Cheng-Ting Chien (the director of NPAS at Academia Sinica, ctchien@gate.sinica.edu.tw) recently informed me that his manuscript would be submitted to the *OPEN BIOLOGY* at the end of November, a little later than expected. Still, we would like to publish the four review articles from NPAS together in the same issue in early 2019. Thank you so much for your help.

We would like to thank you and both reviewers again for their constructive critiques. We hope that you and the reviewers will now find our paper acceptable by the *OPEN BIOLOGY*.

Sincerely,

Yijuang Chern, Ph. D.
Distinguished Research Fellow
Institute of Biomedical Sciences,
Academia Sinica
Taipei, Taiwan
TEL: 886-2-26523913
Email: bmychern@ibms.sinica.edu.tw